# The Effects of Low-Energy Moderate-Carbohydrate (MCD) and Mixed (MixD) Diets on Serum Lipid Profiles and Body Composition in Middle-Aged Men: A Randomized Controlled Parallel-Group Clinical Trial

**DOI:** 10.3390/ijerph17041332

**Published:** 2020-02-19

**Authors:** Małgorzata Magdalena Michalczyk, Adam Maszczyk, Petr Stastny

**Affiliations:** 1Institute of Sport Science, The Jerzy Kukuczka Academy of Physical Education, 40-065 Katowice, Mikolowska 72a, Poland; a.maszczyk@awf.katowice.pl; 2Faculty of Physical Education and Sport, Charles University in Prague, 162 52 Prague, Czech Republic; stastny@ftvs.cuni.cz

**Keywords:** carbohydrate-restricted diet, low energy, weight loss, cholesterol, body fat

## Abstract

Carbohydrate-restricted diets have become very popular due to their numerous health benefits. The aim of this study was to determine the influence of 4 weeks of a well-planned, low-energy moderate-carbohydrate diet (MCD) and a low-energy mixed diet (MixD) on the lipoprotein profile, glucose and C-reactive protein concentrations, body mass, and body composition in middle-aged males. Sixty middle-aged males were randomly assigned to the following groups: hypocaloric MCD (32% carbohydrates, 28% proteins, and 40% fat), hypocaloric MixD (50% carbohydrates, 20% proteins, and 30% fat), and a conventional (control) diet (CD; 48% carbohydrates, 15% proteins, and 37% fat). The participants who were classified into the MCD and MixD groups consumed 20% fewer calories daily than the total daily energy expenditure (TDEE). Baseline and postintervention fasting triacylglycerol (TG), LDL (low-density lipoprotein) cholesterol (LDL-C), HDL (high-density lipoprotein) cholesterol (HDL-C), total cholesterol (tCh), glucose (Gl), and C-reactive protein were evaluated. Body mass (BM) and body composition changes, including body fat (BF), % body fat (PBF), and muscle mass (MM), were monitored. Compared with MixD and CD, MCD significantly changed the fasting serum concentrations of TG (*p* < 0.05), HDL-C (*p* < 0.05), LDL-C (*p* < 0.05), tCh (*p* < 0.05), and glucose (*p* < 0.01). Additionally, body fat content (kg and %) was significantly reduced (*p* < 0.05) after MCD compared with MixD and CD. After the MixD intervention, BM and MM decreased (*p* < 0.05) compared with baseline values. Compared with baseline, after the MixD, BM, MM, tCh, LDL-C, and TG changed significantly. The 4 week low-energy MCD intervention changed lipoproteins, glucose, and body fat to a greater extent than the low-energy MixD. A hypocaloric MCD may be suggested for middle-aged male subjects who want to lose weight by reducing body fat content without compromising muscle mass.

## 1. Introduction

Unhealthy adult lifestyles, especially improper dietary habits, are believed to play an important role in the development of contemporary-civilization diseases [1,2,3]. Numerous studies have investigated the association between diet and obesity, cardiovascular diseases, diabetes mellitus, hypertension, atherosclerosis, and many other diseases [3,4,5,6]. In the diet of modern Europe, the carbohydrate content ranges between 31.9% and 63.2% of all calories consumed, while the fat content ranges between 30% and 40% [7,8]. Unfortunately, it is not so much the amount of carbohydrates and fats as the type of these nutrients that leads to health risks [4,5,6,7,8]. Quite often, the basic diets of Europeans are far from the recommendations of national institutes [5,7,8]. A significant amount of carbohydrates consumed are easily digestible sugars, while the amount of ingested fiber very often does not reach the appropriate level [5,6,7]. The type of fats that Europeans eat is also far from the recommendations of a healthy diet [4,5,6]. 

For years, scientists, dieticians, and physicians have been searching for a diet that would support treatment and prevent diet-dependent diseases [2,3,4,6,9,10]. Specialists who work with patients suffering from obesity, atherosclerosis, or diabetes mellitus have experimented with different dietary procedures [3,4,10,11]. At the end of the 20th century, when the number of adults with obesity in the world increased significantly, low-calorie diets were the most often used to reduce body weight and body fat content [3,6,12,13,14]. Those studies confirmed the positive influence of calorie-restricted diets on body mass reduction, blood pressure, glucose metabolism, lipid profile, and immune response [3,6,12,14]. Reducing the amount of calories in the diet by approximately 20–30% can lead to significant health benefits [3,6]. Unfortunately, calorie-restricted diets are very strict, and patients do not want to stick to them for a long time [7,15,16]. The main disadvantage of such diets is the drastic reduction of daily calorie intake. As a result, after 2–3 weeks of use, patients give up the diet and return to their previous habits [7]. Patients explain this by excessive hunger and irritability and even a worse mood [7,15,16]. Additionally, even with drastic calorie reduction, the results are less than expected [15,16]. Unfortunately, they feel failure and resignation and lose their motivation, but motivation is a critical component of continued obesity treatment [7,17]. That is why we are still looking for the perfect diet model that would not be too restrictive and would yield results in the form of body fat reduction. 

In the last few years, scientists have found that not only the caloric content of the diet but also its composition is important in the process of weight reduction or prevention of diet-related disease [4,9,10,11]. The advantage of low-carbohydrate diets compared with mixed low-calorie diets is the lack of hunger, the rapid reduction of body fat, and the significant impact on blood lipid profile parameters or glucose levels [9,11]. Most scientists confirm that excessive carbohydrate consumption with limited daily energy expenditure could have unfavorable consequences on body fat content, glucose and insulin levels, and the blood lipid profile [2,4,9,11]. Conversely, it has been reported that carbohydrate-restricted diets reduce body fat; improve serum total cholesterol, triglycerides (TG) and high-density lipoprotein (HDL) and low-density lipoprotein (LDL) cholesterol; and reduce insulin and glucose concentrations [11,18,19]. All of the abovementioned variables are well-recognized risk factors for diabetes and cardiovascular diseases [5,6,11]. Many studies have confirmed that carbohydrate restriction diets have a positive effect on weight loss and body composition [4,9,11,19]. Low or moderate carbohydrate consumption is replaced by higher fat and protein consumption, which suppresses appetite by inhibiting gastric emptying, which leads to a long-lasting satiety following a meal [5]. Appetite inhibition may also be caused by the less tasty meals [4,5]. Changes in hormones such as ghrelin, leptin, or peptide YY during low-carbohydrate dieting can modulate appetite [7,19]. Furthermore, higher protein consumption increases postprandial energy expenditure [5].

In carbohydrate-restricted diets, the main macronutrient consumed is fat, mainly mono and polyunsaturated fatty acids (MUFAs and PUFAs) [4,9,11]. In contrast, a typical Western diet rich in saturated fatty acids (SFA) and carbohydrates, especially simple sugars, and poor in fiber increases body fat content and worsens the blood lipid profile [9,20]. High consumption of SFA increases LDL concentration (LDL-C) and has a direct effect on blood clotting, which has a strong atherosclerotic and carcinogenic effect [6]. In contrast to SFAs, MUFAs and PUFAs improve insulin sensitivity, decrease systolic blood pressure, increase HDL concentration (HDL-C), decrease serum TG and LDL-C, inhibit lipogenesis and fat oxidation, and stimulate the immune system [20,21]. MUFAs are also significant sources of energy [4,11,21].

There is a lack of studies comparing the influence of a low-energy mixed diet and a low-energy moderate-carbohydrate diet on body mass, body fat, and blood biochemical parameters. The aim of our study was to establish which diet, a low-energy moderate-carbohydrate diet (MCD) or a low-energy mixed diet (MixD), is better at reducing body mass, fat, and muscle mass and positively influences blood lipoproteins, glucose, and CRP in overweight and obese middle-aged men.

## 2. Methods 

### 2.1. Subjects

Sixty middle-aged men who agreed to participate in this study were randomly assigned to the moderate-carbohydrate diet—MCD (*n* = 20), mixed diet—MixD (*n* = 20), and conventional diet—CD (*n* = 20) groups (Figure 1). For randomization, we used the sealed-envelope method. Specifically, during the first visit to the laboratory, each participant drew an envelope with the prescribed diet. The participants who were classified into the MCD and MixD groups consumed 20% fewer calories daily than the total daily energy expenditure (TDEE) and had different macronutrient compositions. The study blinding was not possible due to the need of proper participant preparation for the dietary program. None of them had experience with reductive diets before this study. The participants in the CD consumed the same diet as before the experiment. For the purpose of the study, they did not change the macronutrient proportions or energy consumption. Patients were recruited from a group of people who independently reported to the Dobry Dietetyk diet clinic at the Academy of Physical Education in Katowice. The clinic has existed since 2009 and is run by qualified dietitians. The inclusion criteria were residence in Katowice, age between 40 and 50 y, body fat content up to 30%, body mass index (BMI) up to 25, no use of any kind of diet or food elimination during the last 12 months, and exercising more than two times a week with high intensity. The exclusion criteria were as follows: the intake of any supplements with established lipid and glucose profile; energy expenditure by physical activity >1000 kcal/week; hypercholesterolemia (total cholesterol > 8.0 mM or dyslipidemia therapy); diabetes (glucose > 126 mg/dL or diabetes treatment); hypertension (systolic blood pressure > 140 mmHg and/or diastolic blood pressure > 90 mmHg or antihypertensive treatment); multiple allergies; celiac disease or other intestinal diseases; any condition that could limit the mobility of the subject, making laboratory visits difficult; life-threatening diseases or conditions that could worsen adherence to the measurements or treatments; vegetarianism or the need for other specific diets; and alcoholism or other drug addiction. In the end, 55 participants completed the study. Two subjects from the MCD group and three from the MixD group resigned from the study (Figure 1). These individuals were not able to maintain calorie-restricted diets and consumed fast foods, sweets, and alcohol, which were not included in the prescribed diets.

Before the experiment began, all participants were informed about the study objectives and the accompanying risks and benefits. They were also informed about the possibility of withdrawing from the experiment at any time. All participants read and signed the informed consent to participate in the study. The testing procedures were approved by the Ethics Committee of the Academy of Physical Education in Katowice. 

### 2.2. Dietary Procedures

The dietary intervention lasted 4 weeks. Two individual hypocaloric diet models, MCD and MixD, were used. In both diets, the daily caloric consumption was 20% fewer calories than the total daily energy expenditure (TDEE). The TDEE was calculated according to the commonly accepted model: (TDEE = activity factor × resting metabolic rate) [22]. Resting metabolic rate was measured at the beginning of the experiment by means of a metabolic cart, MetaLyzer 3B (Cortex, Leipzig, Germany), in the Human Performance Laboratory in the Academy of Physical Education in Katowice. Activity factor was determined based on available indicators for low-physical-activity adults (Activity factor = 1.4) [22]. Additionally, before the experiment, the subjects were asked to take home and complete the 72 h food diary (two weekdays and one weekend day). The dietary records were estimated by a nutritionist to assess previous feeding habits and daily caloric consumption. Before the experiment, all participants consumed a typical Western-style diet [5,6,19]. This type of diet is characterized by high intake of saturated fat, simple sugars, and salt. The respondents consumed fast food, alcohol, sweetened beverages, and sweets. The diet consumed by all participants before the experiment was composed of ~50% carbohydrates, ~15% protein, and ~35% fat. They had never experimented with calorie-reductive diets or with low-carbohydrate diets before.

### 2.3. Diet Composition 

During the experiment, the participants consumed the MCD, MixD, or CD. The composition of the diets is shown in Table 1, and one-day examples in Appendix A. 

The MCD was composed of 32% carbohydrates, 28% proteins, and 40% fat. This diet was composed in such a way that MUFAs and PUFAs constituted 20% and 15% of the total energy intake (TEI), respectively. The MCD meals consisted of poultry, fish, beef, veal and lamb, dried beef, chopped meet tartare, carpaccio and cured ham; olive oil, butter, green vegetables without restriction (raw and cooked), boiled eggs, and seasoned cheese (e.g., mozzarella, halloumi). Warm drinks were restricted to tea and coffee without sugar or herbal extracts. The foods and drinks that participants avoided included alcohol and any sweets such as sugar. They also did not consume white bread, pasta, white rice, sweet milk, fruit yogurt, sweets, soluble tea, or barley coffee. 

The MixD was composed of 50% carbohydrates, 20% proteins, and 30% fat. Saturated fatty acids (SFAs) made up 8%–10%, MUFAs 10%–12%, and PUFAs 10% of the calories (Table 2). In this diet, the consumed carbohydrates came from products with a low or medium glycemic index, such as whole-grain bread and pasta, graham rolls, whole-grain rice, legumes, raw vegetables, poultry, beef, pork, and fish. The participants consumed medium-glycemic-index fruits such as strawberries, grapefruits, and oranges.

The control group consumed their usual diet, named the conventional diet (CD), which was very similar to the modern Western diet [5,19]. The Western-style diet is characterized by high caloric intake of energy-dense foods, saturated and n-6 fatty acids, and refined sugars; excessive salt and alcohol intake; and low consumption of n-3 fatty acids and fiber [5,7]. The Western diet is based on fast food, sweetened beverages, sweets, and alcohol [7]. The CD was composed of 48% carbohydrates, 15% proteins, and 37% fat. SFAs made up 15%, MUFAs 6%, and PUFAs 14% of the calories (Table 2). The diet contained mainly white flour products (bread, bagel, and pasta), white rice, potatoes, beef, pork, sausages, poultry, cheese, fried eggs, vegetables, fruits, margarine, sunflower oil, whole milk, coffee with milk and sugar, tea, fruit juices, carbonated drinks such as cola, and water. The participants consumed medium- and high-glycemic-index snacks such as buns, chocolate bars, bananas, and meals with white rice, potatoes, boiled carrots, and beetroots. 

The meals were prepared in the form of 24 h menus for all seven days of the week—four main meals and one snack a day. The particular diet composition was analyzed using DIETETYK 6.0 software (Jumar, Poznan, Poland).

Our study was unique in the sense that the MCD and MixD were composed of high-quality food products. In the MCD group, the subjects consumed healthy fats, mainly monounsaturated fatty acids from olive oil, dairy products, and nuts, which accounted for more than 50% of all fatty acids consumed. The MCD also contained n-6 and n-3 polyunsaturated fatty acids in a ratio of 4:1. The diet included the consumption of fish, such as mackerel, herring, and sardines, which are rich in n-3 fatty acids. Additionally, the MCD included high-quality protein products such as fish, meat, eggs, and dairy products. In the MixD protocol, the participants ate healthy carbohydrates such as cereals, rice, buckwheat, millet, and fruits. They also ate high-quality protein and fat products similar to those of the MCD. In both diets, the subjects did not eat processed carbohydrates—fast foods, sweets, and carbonated drinks.

### 2.4. Diet Control

In both diets, MCD and MixD, all meals were well planned. The quality and quantity of the food products used to prepare the meals was strictly specified to maintain proper proportions between the major macronutrients. During the 4 weeks of the experiment, the participants consumed meals prepared by the dietetic catering company Catering Service, Powstańców St. 1/1, 41–500 Chorzów. The cooks were given a detailed list of products with a weight and a recipe for preparation. Portioned meals in boxes for the day were delivered by the caterer to every home. Although the subjects did not prepare the meals themselves, before starting the experiment, during the individual visits to experienced dietitians, they were informed what proportions would be given in their diets and which products would dominate the meals. Each of them was given a detailed list containing the food permitted and prohibited in the MCD and MixD. This knowledge increased their opportunity to control their diet compositions. They were also asked to control the weight of the chosen products in their meals at least once a week. The participants took pictures of consumed meals and analyzed them during follow-up visits after 2 weeks of the experiment.

### 2.5. Experimental Design

Before and after four weeks of MCD, MixD, and CD, fasting blood evaluations and somatic measures were carried out to determine several anthropometric and biochemical variables.

### 2.6. Biochemical Analysis

Before and after MCD, MixD, and CD, the following biochemical variables were evaluated in all study participants: triglycerides (TG, mg/dl), total cholesterol (tCh, ng/dl), high-density lipoprotein cholesterol (HDL-C, mg/dl), low-density lipoprotein cholesterol (LDL-C, ng/dl), and glucose (Gl, mg/dl) using Randox UK diagnostic kits (TRIGS-210, CHOL-201, HDL-2652, LDL-2656, Ranbut, Gluc-PAP; Randox Laboratories Ltd. London, UK). C-reactive protein concentration was assessed by an immune test using a CRPL2 Cobas Integra 400/800 diagnostic kit from Roche (Roche Diagnostics GmbH, Mannheim, Germany.).

### 2.7. Body Composition Measurement

The analysis of body mass (BM), body fat (BF), muscle mass (MM), and total body water (TBW) was done using the bioelectrical impedance method with an eight-electrode system (InBody 720, Biospace Co., Tokyo, Japan). Before each measurement, the following testing procedures were maintained: the last meal preceding the body composition evaluations was consumed at 20:00, and then the subjects ingested 1 L of medium-mineralized water. The subjects did not drink alcohol 48 hours before the test, and immediately before the measurement, they emptied their bladders. 

### 2.8. Statistical Analysis

Age, body mass, body composition, and biochemical variables are expressed as mean ± SD. Before using a parametric test, the assumption of normality was verified using the Kolmogorov–Smirnov test. One-way factorial ANOVA was used with significance set at *p* < 0.05. When appropriate, a Tukey post hoc test was used to compare selected data, and the effect size (eta-squared; η^2^) of each test was calculated to determine the significance of the results. The effect size was classified according to Hopkins as 0.01—small, 0.06—medium, and 0.14—large [23]. The remaining analyses were performed using STATISTICA (Stat Soft, Inc. (2018) version 13, Tulsa, OK 74104, USA).

## 3. Results

Sixty middle-aged men (age 45.8 ± 4.5 y; body mass (BM) 107.51 ± 6.23 kg; body height 184.8 ± 8.6 cm; body fat content (%BF) 32.86% ± 4.51%, muscle mass (MM) 42.50 ± 2.01 (kg) were included. In Table 2 and Table 3 and in Figure 2 and Figure 3, the intragroup and intergroup baseline and postintervention results of lipoprotein profile, Gl and CRP concentration, body mass, and body composition values are presented. 

### 3.1. The Biochemical Parameter Changes

Intragroup differences in the postintervention tCh, TG, LDL-C, HDL-C, and Gl values were observed compared with baseline values in the MCD group. Lower postintervention concentrations of TG (F = 18.71, *p* = 0.001, η^2^ = 0.17), LDL-C (F = 32.34, *p* = 0.001, η^2^ = 0.16), tCh (F = 14.90, *p* = 0.001, η^2^ = 0.10), and Gl (F = 45.41, *p* = 0.021, η^2^ = 0.03) (Table 2) and higher postintervention concentrations of HDL-C (F = 13.69, *p*= 0.001, η^2^ = 0.09) were observed (Table 2 and Figure 2).

Additionally, in the MixD group, postintervention tCh, LDL-C, and TG values were different from baseline values. Lower postintervention concentrations of tCh (F = 28.21, *p* = 0.030, η^2^ = 0.04), LDL-C (F = 22.24, *p* = 0.031, η^2^ = 0.05), and TG (F = 34.30, *p* = 0.001, η^2^ = 0.07) were observed.

Lower postintervention values of tCh (F = 33.31, *p* = 0.034, η^2^ = 0.04), TG (F = 36.11, *p* = 0.021, η^2^ = 0.08), HDL-C (F = 27.71, *p* = 0.041, η^2^ = 0.04), and LDL-C (F = 26.61, *p* = 0.013, η^2^ = 0.05) were observed after the MCD diet compared with the postintervention values of tCh, TG, HDL-C, and LDL-C in the MixD and CD groups. Additionally, lower postintervention Gl (F = 44.61, *p* = 0.023, η^2^ = 0.03) was seen in the MCD group compared with the CD group. No differences were observed in tCh, HDL-C, LDL-C, TG, Gl, or CRP concentrations between the MixD and CD groups (Table 2).

### 3.2. Body Composition Changes 

Effects of the dietary interventions on BM, BF (%, kg), and MM in intragroup measures were revealed. A decrease in postintervention values of BM (F = 62.112, *p* = 0.001, η^2^ = 0.06), BF (kg) (F = 49.352, *p* = 0.001, η^2^ = 0.07), and BF (%) (F = 851.252, *p* = 0.001, η^2^ = 0.14) after MCD. Similarly, after the MixD diet, decreases in BM (F = 56.132, *p* = 0.001, η^2^ = 0.07), and MM postintervention values (F = 48.132, *p* = 0.001, η^2^ = 0.06) (Table 3) were revealed.

Lower values of BF (kg) were registered after the MCD (F = 8.32, *p* = 0.004, η^2^ = 0.08) as well as BF (%) (F = 5.46, *p* = 0.007, η^2^ = 0.15) compared with the BF (kg) after the MixD and CD (Table 3 and Figure 3). A difference in MM after the MCD compared with MixD (F = 55.152, *p* = 0.03, η^2^ = 0.04) was also observed.

## 4. Discussions

Excessive adipose tissue and lipoprotein disorders are substantial factors in the development of atherosclerosis, obesity, insulin resistance, and other diseases [5,6,24,25,26]. In particular, elevated LDL-C, which is a strongly prominent factor in arteriosclerosis, will be oxidized, and this increases atherosclerotic changes in blood vessels [5,9,11]. In the past decade or two, numerous studies have been carried out on the impact of two diet models, low-calorie and low-carbohydrate diets, to reduce fat content and decrease the risk of dyslipidemia or insulin resistance in overweight and obese adults [3,6,9,11,18,19,24]. Many researchers have shown that low-carbohydrate diets have a positive effect on triglyceride levels [9,18,19,25,26,27,28], adipose tissue [9], blood glucose, and CRP concentration [9,26]. The results of these studies allow researchers to propose this diet as a solution to the rapidly progressing problems of overweight and obesity, as well as lipoprotein diseases in adults [26]. Recently, the World Health Organization (WHO) has warned overweight and obese people, as well as those with diabetes, to reduce their daily carbohydrate consumption [24]. 

Our subjects were middle-aged men with lipoprotein imbalance and high levels of body fat. In our experiment, we evaluated the effects of two different diet models: a moderate-carbohydrate diet (MCD) with 20% fewer calories than TDEE and a mixed diet (MixD) also with 20% calorie restriction. The MCD contained 40% fat, 32% carbohydrates, and 28% protein. The MixD contained 50% carbohydrates, 30% fat, and 20% protein. We included a control group that consumed 48% carbohydrates, 35% fat, and 15% protein. We wanted to learn how diets with different macronutrient compositions and the same calorie reduction would influence the lipoprotein profile and body composition in middle-aged men with lipoprotein disorders and high body fat content. 

Despite our previous positive experience with a carbohydrate restriction diet—the low-carbohydrate diet and even the low-carbohydrate ketogenic diet—on lipid profile and body composition [4,9], in this study we decided to experiment with a moderate-carbohydrate diet that contained relatively higher amounts of carbohydrates, at 25%–45%, to compare with the low-carbohydrate and the ketogenic diets that contain 10%–25% and <10% carbohydrates, respectively. In our earlier studies, the participants included well-motivated athletes determined to achieve better body mass and composition to improve their performance. Unfortunately, in ordinary people, such as our responders, who consume significant amounts of carbohydrates, especially snacks, fizzy drinks, and alcohol, strict carbohydrate reduction is very difficult [5,6]. They declare that both carbohydrate products and alcohol relax them, and they give them up reluctantly, despite their negative influence on health [5,6]. Most of them do not want to give up sweets, fast foods, and alcohol, and they are aware that these products are unhealthy [5,6,9]. In the past, we also experimented with a reduced-calorie diet [3,6]. After six weeks of diet with reduced calories, we did not observe as many changes in body fat content or in other biochemical parameters as we expected [3,6]. In the diet with daily reduced calories, more than 20% from TDEE, it was very difficult for participants to restrict their sweets and alcohol, and twenty-one out of ninety-four of them left the study [6]. Similar results were observed in this study. After 2 weeks of the study, 5 subjects left, explaining that they were not able to maintain carbohydrate and calorie restriction. These results led us to make a comparative study of both model diets. 

The results indicate that compared with the subjects from the MixD and CD groups, the men who consumed the MCD had significantly lower tCh, TG, and LDL-C and significantly higher HDL-C concentrations. Values of these variables after the MCD and the MixD compared with the values recorded at baseline were significantly different. After MCD, tCh, LDL-C, TG, and Gl were significantly lower, while the HDL-C concentration was higher. After MixD, a significant decrease in tCh, LDL, and TG was recorded. These results confirm that both diets positively affected the lipid profile [4,9,11,19], although a more positive effect was observed after MCD. The significant increase in HDL-C observed after MCD indicates that this diet can be recommended for people with a low HDL-C concentration. Similar results after a carbohydrate-restricted diet were presented by Sharman et al. [18]. Additionally, Rajaie et al. [24], after six weeks of a moderate-carbohydrate diet, observed a tendency toward lower TG and higher HDL-C. These positive effects on the lipid profile and glucose concentration on the MCD could have been caused by both the high quality of consumed fats and the low carbohydrate consumption. In this group, participants consumed mainly MUFAs from olive oil, dairy products, and nuts, as well as n-3 PUFAs from fish. It is known that both n-3 and MUFAs have a positive effect on the blood lipid profile [20,21,28]. 

Volek’s group compared a low-carbohydrate diet (LCD) with a low-fat diet (LFD) and found that after 3 months, tCh and TG were lower in the LCD group than in the LFD group [11]. After four weeks of MCD, subjects significantly decreased their level of LDL-C, which is classified as a strong arteriosclerotic factor. This can be considered a positive effect of the applied carbohydrate-restricted diet compared with the general recommendations of much higher consumption of carbohydrates [4,6,11]. Similarly, Brinkworth et al. [19], after a few weeks of giving a low-carbohydrate diet to obese people, observed a significant decrease in TG and LDL and an increase in HDL compared with the values on a low-fat diet. Additionally, Tay et al. [28] and Paniagua [29], after a low-carbohydrate diet, which was high in monounsaturated fats and low in saturated fat, achieved greater improvements in the lipid profile, exemplified by higher HDL and lower LDL and TG. On the other hand, experimentation with a diet high in MUFAs resulted in an improved LDL/HDL ratio. These results are supported by the research of Paoli et al. (2011) [27], who used a low-carbohydrate diet with a high content of MUFAs and observed decreases in LDL lipoprotein fractions. The authors explained this phenomenon by low amounts of saturated fats and a high content of unsaturated fats, which inhibit HMG reductase activity, an enzyme that controls cholesterol synthesis [30]. 

So far, there are no studies in which the authors compared the effect of a moderately carbohydrate reduced diet on body composition. Excess body fat is not the result of inappropriate eating habits in the last several weeks but improper nutrition in recent years [5]. Scientists, doctors, and dietitians have still not managed to develop a diet model that would, without extreme caloric restriction and macronutrient changes, gradually and safely affect the reduction of body fat. In the last decade, scientists have mainly focused on the effect of low-carbohydrate and ketogenic diets on body composition [4,9,11,19,27]. Very often, despite much scientific evidence about their positive effects, they are not practiced by dietitians because they are too restricted. Dietitians complain that even if they recommend this kind of diet, their patients do not want to follow it for a long time and very often stop the diet after a few weeks. Extreme reduction of carbohydrate consumption among overweight and obese patients is associated with large restrictions and no tasty meals [9]. The purpose of our experiment was to learn whether a moderate-carbohydrate diet with caloric restriction, which was easy to apply because it would not involve significant caloric or carbohydrate restrictions, would have the same positive effects on body composition as diets with drastic restrictions. In our experiment, the MCD significantly affected body fat in the studied middle-aged men. The benefits of the MCD were revealed in absolute (kg) as well as relative (%) reductions of body fat. So far, there are no studies with results we can compare ours to. Our results confirm the concept that the level of body fat does not depend on the amount of fat consumed in the diet but rather on the amount and quality of carbohydrates ingested [9,11]. This thesis explains the paradox of today’s Western-style diet consumed by adults living in highly developed countries. In the latter part of the 20th century, scientists, doctors, and dietitians have encouraged obese adults to consume rather low-fat diets, with light products, i.e., in which fats have been replaced with starch and other sugars [22]. These recommendations have led to the state in which 20% of European residents from well-developed countries are obese [31]. The decrease in BF following the MCD procedure may be related to decreased carbohydrate consumption, which induced lower blood glucose concentrations [9]. After a meal reach in carbohydrates, the level of glucose rises, which immediately stimulates pancreatic insulin secretion [32]. Insulin subsequently binds to its transmembrane tyrosine kinase receptors, located in different tissues, especially in adipose tissue. When activated, insulin receptor starts a cascade reaction through which the protein glucose transporter GLUT4, located in plasma, is translocated into membranes, causing glucose transport across the membrane to the plasma and thereby stimulating lipogenesis [14]. The reduced consumption of carbohydrates during MCD decreases the rate of lipogenesis, which was evidenced in the decrease in body fat content [9,28]. In our earlier experiments, we have evaluated the positive influence of the MCD and MixD on body mass and seen no negative muscle mass changes [4,9]. In the studied group, a significant reduction in body mass and no muscle mass reduction after MCD were observed, while a significant decrease in these variables was observed after MixD [7]. This phenomenon can be explained by the fact that the participants in the MixD group consumed less protein per kg/d compared with the MCD group. In the MixD group, the average protein consumption was 0.96 g/kg/body mass per day, which accounted for approximately 20% of the total daily calorie intake, compared with the MCD group, where it reached 1.42 g/kg/body mass per day, which accounted for 28% of the total daily calorie intake. Higher protein consumption leads to catabolism of muscle tissue [5]. During the body fat reduction process, it is extremely important not to reduce muscle mass, which is directly influenced by the basal metabolic rate [33]. Considering that both dietary protocols were hypocaloric, a higher intake of protein during the MCD intervention most likely inhibited muscle catabolism [34]. 

The factor that probably had a significant impact on the divergent results between our paper and others was that our subjects consumed high-quality products, both on the 4 week MCD and on the MixD, which is an innovative solution in this type of research [4,9]. The meals in the MCD and the MixD procedures were composed of high-quality protein with a full amino acid profile. The participants consumed beef, poultry, eggs, and fish. Additionally, they ate soy protein and other vegetable proteins. The MUFAs and PUFAs were derived from olive oil, nuts, avocado, and several types of seeds [4,9]. During the experiment, the subjects mostly avoided SFA; thus, they limited the consumption of pork, bacon, ham and sausages, liver, yellow cheese, and other deep-fried products [4,9]. In practice, as dietitians, we have observed that many people who want to reduce body mass choose carbohydrate-restricted diets to reduce body fat but consume excess amounts of saturated fats due to the low cost and great access to such food products. They also experiment with extremely low calorie diets, such as 1000–1200 kcal/day, in which all macronutrients are restricted [34]. Insufficient energy consumption induces muscle tissue catabolism rather than fat tissue loss [5,9]. 

It seems logical that limiting the intake of carbohydrates decreases glucose concentrations [9,32]. Our hypothesis stating that MCD will significantly reduce the level of glucose was confirmed [32]. Sharman et al. [18] found that very low carbohydrates also reduced fasting glucose in overweight men. Blood glucose concentration is highly dependent on the level of carbohydrates consumed in the diet and on the insulin glucose transport mechanism. Low levels of n-3 and high n-6 in phospholipids of cellular muscle cell membranes are associated with an increase in their resistance to insulin, which promotes the development of insulin resistance and type II diabetes [20,21]. In MCD, n-3 PUFA consumption was high, which might have positively influenced this mechanism. The obtained results, especially after MCD, support another conclusion, that a diet with a limited amount of carbohydrates has a positive health effect on glucose metabolism and can be recommended to people with elevated levels of glucose [9,32].

Adipose tissue is an active endocrine organ that releases adipokines and cytokines, which can potentially contribute to atherogenesis, coronary artery disease, and metabolic syndrome [35]. Consumption of a low-fiber, high-fat, Western-style diet induces adiposity and adipose inflammation characterized by elevations in the M1:M2 macrophage ratio and pro-inflammatory TNF-α and IL-6 expression [36]. Increased cytokines and adipokines, such as adiponectin, in epicardial and subcutaneous adipose tissues and in serum directly promote coronary artery disease [35]. Koop et al. [37] observed that weight loss reduction by hypocaloric diets reduced CRP levels in obese men. Ridker et al. [38] suggested that elevated plasma levels of CRP have become one of the strongest independent predictors of cardiovascular disease. In our study, we did not evaluate specific cytokine concentration but only C-reactive protein. CRP is released from many organs, such as the liver, adipose tissue, and arterial vessels. CRP blood concentration increases during injury, autoimmune disease, and heart attack but also during bacterial, viral, and fungal infections [38]. We observed that after 4 weeks of MCD and MixD, lower CRP concentrations were recorded, while in the CD group, CRP levels did not change. Similar results were presented by Visser et al. [39], who explained that a lack of significant changes in CRP concentration was related to the fact that circulating plasma CRP is elevated in obese subjects, and its level is also directly correlated with the amount of body fat [35,39]. CRP is not merely an inflammatory marker but directly participates in the process of atherogenesis by modulating endothelial function [38]. In general, our results will be more relevant using the cross-over design and longer period of intervention, which should be applied in future studies. 

## 5. Conclusions

Many physicians and nutrition specialists do not recommend carbohydrate-restricted diets to improve the lipoprotein profile and reduce body mass and fat content. They do not consider the proportion of macronutrients in the diet but focus on reducing the daily caloric content. This is the reason why they recommend low-calorie mixed diets instead of low-carbohydrate diets. Regardless of the daily caloric content of the diet, they always suggest the same standard proportion of macronutrients as in the mixed diet recommendations. In our study, we used two different diet models to evaluate their influence on the lipoprotein profile, as well as body mass and fat reduction. After 4 weeks of the standard mixed diet, MixD, with 20% fewer calories than TDEE, and a moderate-carbohydrate diet, MCD, also with 20% calorie reduction, we found significant positive changes in fasting blood triacylglycerols, HDL-C, LDL-C, tCh, and glucose after the MCD compared with the MixD. In addition, a significantly greater decrease in fat mass and no significant changes in muscle mass after 4 weeks of the MCD were recorded compared with 4 weeks on the MixD. We can conclude that hypocaloric MCD is more effective than hypocaloric MixD in improving the lipid profile and reducing body fat content. 

## Figures and Tables

**Figure 1 ijerph-17-01332-f001:**
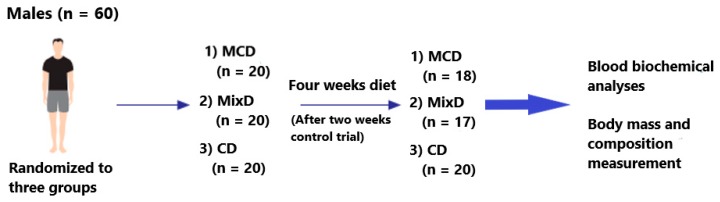
Scheme of the descriptive analysis. MCD — low-energy moderate-carbohydrate diet; MixD — low-energy mixed diet; CD — conventional diet.

**Figure 2 ijerph-17-01332-f002:**
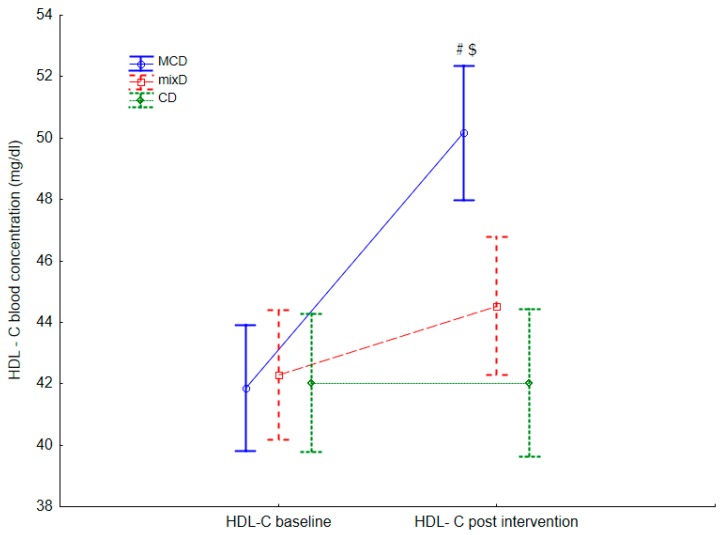
Differences in HDL-C postintervention between the moderate-carbohydrate diet (MCD). Values are expressed as mean and standard deviation. Mix diet (MixD) and conventional diet (CD) groups; #—significant difference compared with base line, $—significant differences compared with the MixD and CD groups, respectively (*p* < 0.05).

**Figure 3 ijerph-17-01332-f003:**
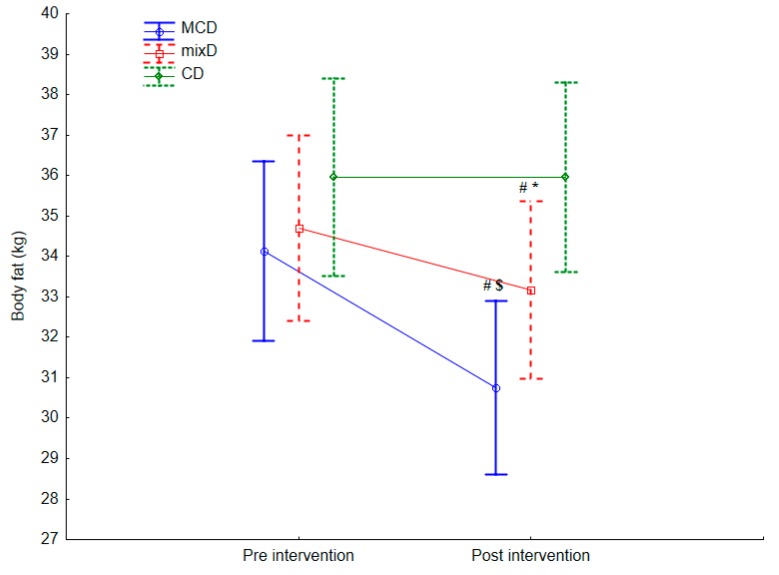
Differences in body fat weight (kg) postintervention between the moderate-carbohydrate diet (MCD), mix diet (MixD), and conventional diet (CD). Values are expressed as mean and standard deviation. (CD) groups; #—significant difference compared with the baseline, $—significant differences compared with the MixD and CD groups, respectively (*p* < 0.05), *—significant difference compared with the CD groups.

**Table 1 ijerph-17-01332-t001:** Average macronutrients and total energy content of the moderate-carbohydrate diet (MCD), mix diet (MixD), and conventional diet (CD) reported during study.

Contents	MCD Mean ± SD	MixD Mean ± SD	CD Mean ± SD
RMR, kcal	1937.5 ± 47	1955 ± 51	2240 ± 51.2
TEI, kcal	2170 ± 42	2190 ± 53	2800 ± 64
TEI, kJ	9085 ± 175	9169 ± 221	11723 ± 267
Carbohydrate, %	32 ± 1.3	50 ± 1.5	48 ± 4.3
Carbohydrate, g	174 ± 8.6	274±4.1	336 ± 13.4
Fiber, g	24.6 ± 2.6	32.7 ± 1.6	25.6 ± 4.7
Proteins, %	28 ± 1.6	20 ± 1.2	15 ± 3.7
Proteins, g	152 ± 6.2	110 ± 3.2	105 ± 3.9
Proteins g/kg/d	1.42 ± 0.2	0.96 ± 0.3	0.97 ± 0.4
Fat, %	40 ± 1.2	30 ± 0.8	37 ± 3.0
Fat, g	96.5 ± 3.3	73 ± 1.5	115 ± 3.45
SFA, g	10.5 ± 2.4	22 ± 2.1	58±4.3
MUFA, g	58 ± 4.8	27 ± 2.2	22 ± 3.2
PUFA, g	28 ± 2.6	24 ± 1.5	35 ± 5.2
n-6, g	21 ± 1.8	19.5 ± 2.6	33 ± 5.1
n-3, g	7 ± 1.6	4.5 ± 0.2	2 ± 1.4
EPA and DHA, g	4.5 ± 0.7	1.0 ± 0.7	0.5 ± 0.2
n-6/n-3	3:1	4.5:1	17:1
Cholesterol, g	286.8 ± 10.6	272.5 ± 14.4	750.6 ± 35.7

CHO—carbohydrates, PRO—proteins, SFA—saturated fatty acids, MUFA—monounsaturated fatty acids, PUFA—polyunsaturated fatty acids, EPA—eicosapentaenoic acids, DHA—docosahexaenoic acids, n-3—omega 3, n-6—omega 6, TEI—total energy intake, RMR—resting metabolic rate.

**Table 2 ijerph-17-01332-t002:** Changes in biochemical variables after the dietary interventions of the moderate-carbohydrate diet (MCD), mix diet (MixD), and conventional diet (CD).

Variables	MCD Mean ± SD		MixD Mean ± SD		CD Mean ± SD	
	Baseline	Post Intervention	Baseline	Post Intervention	Baseline	Post Intervention
tCh (mg/dl)	247.28 ± 19.91	216.11 ± 20.30 **^,#,$^	256.63 ± 17.91	240.93 ± 15.45 *	250.36 ± 15.17	253.64 ± 20.12
HDL-C (mg/dl)	40.85 ± 3.35	48.16 ± 2.61 **^,#,$^	42.12 ± 4.50	44.37 ± 6.93	41.93 ± 4.41	40.21 ± 5.12
LDL-C (mg/dl)	164.45 ± 12.70	132.0 ± 10.34 **^,#,$^	167.70 ± 11.52	153.50 ± 10.70 *	166.80 ± 12.21	165.20 ± 14.23
TG (mg/dl)	183.41 ±3 5.25	106.40 ± 27.10 **^,#,$^	187.63 ± 32.42	164.81 ± 27.50 *	189.5 ± 33.80	191.43 ± 28.10
Gl (mg/dl)	105.6 ± 5.30	94.30 ± 3.21 *^,$^	106.18 ± 6.34	100.40 ± 5.58	105.15 ± 5.42	106.15 ± 6.42
CRP (mg/l)	8.53 ± 3.72	5.01 ± 2.4	7.87 ± 2.1	6.43 ± 3.2	8.45 ± 3.8	8.61 ± 4.1

Note: tCh—total cholesterol, HDL-C—high-density lipoprotein cholesterol, LDL- C—low-density lipoprotein cholesterol, TG—triglycerides, Gl—glucose, CRP—C-reactive protein, *—significant differences compared with the baseline (*p* < 0.05), **—significant differences compared with the baseline (*p* < 0.01), #—significant differences compared with the MixD group (*p* < 0.05), $—significant differences compared with the CD group (*p* < 0.05).

**Table 3 ijerph-17-01332-t003:** Baseline and postintervention body composition results for moderate-carbohydrate diet (MCD), mix diet (MixD), and conventional diet (CD).

Variables	MCD Mean ± SD		Mix D Mean ± SD		CD Mean ± SD	
	Baseline	Post Intervention	Baseline	Post Intervention	Baseline	Post Intervention
BM (kg)	107.85 ± 7.83	105.2 ± 7.31 **	106.72 ± 5.52	104.2 ± 4.71 **	108.14 ± 5.34	107.8 ± 5.40
BMI	32.5 ± 1.7	31.1 ± 1.6	31.6 ± 1.4	30.72 ± 1.3	31.6 ± 1.4	31.2 ± 1.4
BF (%)	32.87 ± 2.84	29.21 ± 3.43 **^, #, $^	32.00 ± 4.86	31.81 ± 3.06	33.80 ± 3.07	33.53 ± 4.21
BF (kg)	34.97 ± 4.24	30.75 ± 4.69 **^,$^	34.84 ± 4.22	33.27 ± 4.83	36.11 ± 4.41	36.87 ± 4.98
MM (kg)	42.43 ± 2.27	41.85 ± 1.61 ^#^	42.89 ± 2.06	40.85 ± 1.53 **	42.20 ± 1.70	41.76 ± 2.13
TBW (kg)	35.46 ± 2.31	34.14 ± 2.26 ^#^	34.7 ± 2.37	31.6 ± 2.42 **	35.7 ± 2.16	35.1 ± 2.48

Note: BM—body mass, BMI—body mass index, BF—body fat, MM—muscle mass, TBW—total body water. **—significant differences compared with the baseline (*p* < 0.01), #—significant differences compared with the MixD group (*p* < 0.05), $—significant differences compared with the CG group (*p* < 0.05).

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
