# Peer review of "The Effects of Low-Energy Moderate-Carbohydrate (MCD) and Mixed (MixD) Diets on Serum Lipid Profiles and Body Composition in Middle-Aged Men: A Randomized Controlled Parallel-Group Clinical Trial"

_ijerph, 2020, doi:10.3390/ijerph17041332_

Round 1

Reviewer 1 Report

This is a randomized controlled study to compare MCD, MixD, and CD. There are several incomprehensible points.

Major

Why did author analyze with a two-way repeated measure ANOVA? I feel one way factorial ANOVA is better, to compare the changes of parameters in three groups during intervention. Authors described “Significant differences in post intervention of Tch…Gl were observed compared to baseline values in the MCD”. Was this analysis made in a two-way repeated measure ANOVA? Please show ANOVA table in each parameters. In my knowledge, Bonferroni test does not use F-value. But authors described F-value (lines 225-227). Please explain. In this manuscript, authors tried to mention type of fatty acids. But three groups are different with macronutrient balance and energy intake. Thus, I feel this trial did not approach balance of type of fatty acids. Authors should delete the description as for type of fatty acids.

Minor

Many references are not appropriate. For example, although reference 1 must be a review about correlation between lifestyle and diseases, it is a paper about experiments in PGC-1alpha KO mice. At least, 10 of 49 references should be replaced. In lines 40 and 43, authors described “recommendations”. Please refer two or more recommendations or guidelines from scientific organizations. In lines 108-110, authors described subjects’ characteristics. But subjects’ characteristics are results. Please move this description to Results section. As well as characteristics, the number of 55 subjects is also result. Please move it. On the other hand, authors did not describe how to recruit these subjects. Furthermore, I cannot find inclusion criteria. In lines 140, authors described they measured RMR with MetaLyzer 3B. When and where did author measure RMR? According to table 2, MCD group consumed diet with average 2170 kcal and SD 42 kcal. Why was SD so narrow? As well as MCD, SD of MixD and CD were narrow? Please show the RMR data in each group. In lines 141, activity factor was described as “low physical activity adults”. Is it 1.3, 1.5, or 1.7? Because I (and most of readers) cannot read reference 30, please explain it specifically. In lines 142, authors described all participants consumed typical Western style diet. In lines 162, authors described CD group consume usual diet that is very similar to the present Western diet. As a reference of Western diet, authors referred Eur J Clin Nutr 2013, 67, 789-796. But this paper is a review of ketogenic diet. What is typical Western style diet? Please explain the detail. (As mentioned above, many references are not appropriate.) In lines 188, authors describe all experiment meals were prepared by dietetic catering company. Please clarify the name and address of this company. As supplementary information, please show the typical one-day menu and recipe of experiment meals in each group. Figure title were wrong in figure 2-4. HDL in figure 2 should be HDL-C. HDL in figure 3 should be body fat percentage. HDL in figure 4 should be body fat weight. Figure legends were wrong in figure 2-4. CG in figure 2-4 should be CD. I think figure 3 and 4 are redundant. Please remove one of two.

Author Response

This is a randomized controlled study to compare MCD, MixD, and CD. There are several incomprehensible points.

Answer: Thank you for your tine to make review, which improved our manuscript. We believe we resolved  all of your concerns.

 Major

Why did author analyze with a two-way repeated measure ANOVA? I feel one way factorial ANOVA is better, to compare the changes of parameters in three groups during intervention.

Answer: A one way ANOVA was used with significance set at p<0.05. When appropriate, a Bonferroni post hoc test was used to compare selected data, the effect size (eta-squared; η2) of each test was calculated to determine the significance of results. So now the one-way was used, and we recalculated the results to One -way ANOVA. Most of p values are quite low 0.001 it changed just couple numbers.

Authors described “Significant differences in post intervention of Tch…Gl were observed compared to baseline values in the MCD”. Was this analysis made in a two-way repeated measure ANOVA?

Answer: Yes originally one-way was used, but this do not fully explain the analyses between groups post intervention. Now we used the factorial model one way ANOVA.

Please show ANOVA table in each parameters. In my knowledge, Bonferroni test does not use F-value. But authors described F-value (lines 225-227). Please explain.

Answer: Now is one-way ANOVA used only, which we reported in text if significant and showed in table 2 as post hoc result.

In this manuscript, authors tried to mention type of fatty acids. But three groups are different with macronutrient balance and energy intake. Thus, I feel this trial did not approach balance of type of fatty acids. Authors should delete the description as for type of fatty acids.

Answer: The introduction part regarding fatty acids have been shortened.

Minor

Many references are not appropriate. For example, although reference 1 must be a review about correlation between lifestyle and diseases, it is a paper about experiments in PGC-1alpha KO mice. At least, 10 of 49 references should be replaced.

Answer: We doublechecked and corrected the references as you suggested.

In lines 40 and 43, authors described “recommendations”. Please refer two or more recommendations or guidelines from scientific organizations.

Answer: We add two references

EFSA Panel on Dietetic Products, Nutrition and Allergies (NDA Scientific opinion on dietary reference values for carbohydrates and dietary fibre. EFSA J. 2010;8:1462. 

USDA, USDHHS . 2015–2020 Dietary Guidelines for Americans. 8th ed. U.S. Government Printing Office; 2015. [(accessed on 30 November 2018)]. Available online: https://www.cnpp.usda.gov/2015-2020-dietary-guidelines-amaricans.

In lines 108-110, authors described subjects’ characteristics. But subjects’ characteristics are results. Please move this description to Results section.

Answer: We moved subjects characteristic to the results

As well as characteristics, the number of 55 subjects is also result. Please move it. On the other hand, authors did not describe how to recruit these subjects. Furthermore, I cannot find inclusion criteria.

Answer: Recruitment procedure and inclusion criteria paragraphs were added in subject section.

 In lines 140, authors described they measured RMR with MetaLyzer 3B. When and where did author measure RMR?

Answer: The answer was added to the dietary procedure paragraphs.

According to table 2, MCD group consumed diet with average 2170 kcal and SD 42 kcal. Why was SD so narrow? As well as MCD, SD of MixD and CD were narrow?

Answer: In preparing the diet we have tried to make in terms of calorie in this group did not differ significantly, and even the SD for body mass and % of body fat was narrow. Thus from some point of view we actually recruited quite homogenous group.

Please show the RMR data in each group.

Answer: The RMR was added to the table 1

In lines 141, activity factor was described as “low physical activity adults”. Is it 1.3, 1.5, or 1.7?

Answer: Physical activity factor was 1.2, which is now stated.

In lines 142, authors described all participants consumed typical Western style diet. Because I (and most of readers) cannot read reference 30, please explain it specifically.

Answer: This information was added  in dietary procedure paragraphs in method section

In lines 162, authors described CD group consume usual diet that is very similar to the present Western diet. As a reference of Western diet, authors referred Eur J Clin Nutr 2013, 67, 789-796. But this paper is a review of ketogenic diet. What is typical Western style diet? Please explain the detail. (As mentioned above, many references are not appropriate.)

Answer: We added a sentenced about Western style diet and changed references. 

In lines 188, authors describe all experiment meals were prepared by dietetic catering company. Please clarify the name and address of this company.

Answer: The name and address of catering company was added.

As supplementary information, please show the typical one-day menu and recipe of experiment meals in each group.

Answer: The typical one-day menu and recipe were added in supplementary file 1.

Figure title were wrong in figure 2-4. HDL in figure 2 should be HDL-C. HDL in figure 3 should be body fat percentage. HDL in figure 4 should be body fat weight. Figure legends were wrong in figure 2-4. CG in figure 2-4 should be CD. I think figure 3 and 4 are redundant. Please remove one of two.

Answer: We corrected all errors and removed figure 3

Reviewer 2 Report

The study of Michalczyk M.M et al., is a well designed study that compares the beneficial effects of 4-week low energy moderate carbohydrate diet 2 versus 4-week of low energy mixed diet. The design of the study is adequate and they manage to have 3 experimental groups with no differences at baseline. So, the differences reported after the 4-week experimental period is due to the different diets consumed. 

However there are some major comments on the manuscript of the authors.

1. To begin with, there is an essential need for a native English speaker to review the manuscript. There are so many typos and gramatical errors that reduce the value of the manuscript. Just two examples for that:

Line 13. have shown instead of have showed

Line 14. aimed to compare instead of aimed to compere

As I mentioned, there are many more through the manuscript. 

2. In my opinion, there is no need for baseline characteristics to appear in the abstract. 

3. Introduction is well structured but is mainly focused on the effects of type of fats (SFA, MUFA, PUFA) in the carbohydrate restricted diets with no mention of the effects of type of carbohydrate consumed (simple vs complex).

4. Authors need to present the baseline characteristics of the subjects included in the study as a table and show whether there are statistical differences between them. 

5. It is strange why authors do not mention body weight of the subjects and BMI. What about total body water, HOMA-IR? Are there any differences in these parameters? Please explain.

6. Table 1 represents macronutrient content of the experimental diets. However without statistical analysis. Moreover, authors need to explain how these data were obtained. These data reflect the analysis of daily consumed diets of the participants of each group?

7. The major problem in the methodology followed is that MixD and MCD diet not only differ in the amount of carbohydrates includes but in the % of the different types of diet. MCD contains higher % of SFA and lower % of MUFA when compared to MixD. This raises great concerns on which macronutrient finally makes the difference and leads to the observed beneficial effects. 

8. It is clear that the study paid great attention in the control of the subjects and their adaptation in the diet and experimental procedures. However it is not clear whether the subjects of all experimental groups received cooked meals and whether they could consume other meals apart from the ones that were provided. A better explanation could be of a great use. 

9. There is no need to mention results with no statistical differences. No need to repeat "post hoc Bonferroni test showed.....". It makes the text tiring and confusing. Also, I believe that it would improve the value of the manuscript if sections 3.1 and 3.2 were explained in the same section. The same for sections 3.3 and 3.4  etc. Summarizing the results would make them easier to understand. 

11. I believe it would be better if results followed the material and methods order.

12. Authors need to review the abbreviations used. For example they use GL for glucose but then they change it for Gl. What BM, BF, MM stand for? What about LCD and LFD in the discussion section?

13. Figures 2,3,4 have the same figure caption. Please check. 

14. Discussion is really confusing and authors should focus on results that they have tested rather that hypothesize on resutls they did not check. Authors should compare their experimental design or outcomes with of other authors. 

15. Authors mention the beneficial effects of high protein intake and weight loss. However, they do not mention the renal problems high protein intake may cause. 

Author Response

The study of Michalczyk M.M et al., is a well designed study that compares the beneficial effects of 4-week low energy moderate carbohydrate diet 2 versus 4-week of low energy mixed diet. The design of the study is adequate and they manage to have 3 experimental groups with no differences at baseline. So, the differences reported after the 4-week experimental period is due to the different diets consumed. 

However there are some major comments on the manuscript of the authors.

To begin with, there is an essential need for a native English speaker to review the manuscript. There are so many typos and gramatical errors that reduce the value of the manuscript. Just two examples for that:

Answer:  Thank you for time spend on improvement of our manuscript. The native speaker editing has been applied.

Line 13. have shown instead of have showed

Answer: Corrected

Line 14. aimed to compare instead of aimed to compere

Answer: Corrected

As I mentioned, there are many more through the manuscript. 

Answer: The native speaker editing has been used for whole document.

In my opinion, there is no need for baseline characteristics to appear in the abstract. 

Answer: Baseline characteristics in the abstract was deleted

Introduction is well structured but is mainly focused on the effects of type of fats (SFA, MUFA, PUFA) in the carbohydrate restricted diets with no mention of the effects of type of carbohydrate consumed (simple vs complex).

Answer: We agree with the reviewer that this paragraph is written in too much detail. Therefore, it shortened and added information about the effects of excess carbohydrates in the diet on health.

Authors need to present the baseline characteristics of the subjects included in the study as a table and show whether there are statistical differences between them. 

Answer: The baseline characteristic of the subject is in the  table 3 

It is strange why authors do not mention body weight of the subjects and BMI. What about total body water, HOMA-IR? Are there any differences in these parameters? Please explain.

Answer: BMI parameter changes do not show exactly which component of body composition has been reduced. The water level fluctuates easily, as we observed in earlier studies [7], therefore we did not analyzed water changes under the influence of diet. However, according to the reviewer's wishes, we have added BMI and TBW results. The level of fat tissue, expressed in kilograms due to the relatively constant level was for us the most important component in assessing the impact of diets on body composition. We did not measured any hormones so we did not known how HOMA-IR was changed.

Table 1 represents macronutrient content of the experimental diets. However without statistical analysis. Moreover, authors need to explain how these data were obtained. These data reflect the analysis of daily consumed diets of the participants of each group?

Answer: Yes the data in table 1 reflect the analysis of daily consumed diets of the participants of each group, which is now stated in figure caption.

The major problem in the methodology followed is that MixD and MCD diet not only differ in the amount of carbohydrates includes but in the % of the different types of diet. MCD contains higher % of SFA and lower % of MUFA when compared to MixD. This raises great concerns on which macronutrient finally makes the difference and leads to the observed beneficial effects. 

Answer: The reviewer probably looked at the wrong column. In Table 1, the column describing the composition of MCD says that it contained a lower level of SFA and a higher MUFA than MixD.

It is clear that the study paid great attention in the control of the subjects and their adaptation in the diet and experimental procedures. However it is not clear whether the subjects of all experimental groups received cooked meals and whether they could consume other meals apart from the ones that were provided. A better explanation could be of a great use. 

Answer: Paragraph Diet control  was supplemented with the missing information.

There is no need to mention results with no statistical differences. No need to repeat "post hoc Bonferroni test showed.....". It makes the text tiring and confusing.

Also, I believe that it would improve the value of the manuscript if sections 3.1 and 3.2 were explained in the same section. The same for sections 3.3 and 3.4  etc. Summarizing the results would make them easier to understand. 

Answer: All comments were done.

I believe it would be better if results followed the material and methods order.

Answer: Done as suggested.

Authors need to review the abbreviations used. For example they use GL for glucose but then they change it for Gl. What BM, BF, MM stand for? What about LCD and LFD in the discussion section?

Answer: All used  abbreviation was explained.

Figures 2,3,4 have the same figure caption. Please check. 

Answer: The table names have been corrected

Discussion is really confusing and authors should focus on results that they have tested rather that hypothesize on results they did not check. Authors should compare their experimental design or outcomes with of other authors. 

Answer: As the reviewer suggests, we have supplemented the discussion with additional sentences in which we compared our results with those of other authors.

Authors mention the beneficial effects of high protein intake and weight loss. However, they do not mention the renal problems high protein intake may cause. 

Answer: We did not mentioned because our MCD was not high protein diet. Protein consumption in MCD was average on 1.5 g/kg/d what is not very high. European Society for Clinical Nutrition and Metabolism recommend even 1.2g protein/kg bm/d for adults.   Renal problems could appeared in many different reason like low daily water consumption, high sodium consumption bacterial infection or gut microbiota dysbiosis.   

Deutz, N.E.; Bauer, J.M.; Barazzoni, R.; Biolo, G.; Boirie, Y.; Bosy-Westphal, A.; Cederholm, T.; Cruz-Jentoft, A.; Krznariç, Z.; Nair, K.S.; et al. Protein intake and exercise for optimal muscle function with aging: Recommendations from the ESPEN Expert Group. Clin. Nutr. 2014, 33, 929–936. doi:10.1016/j.clnu.2014.04.007.

Valdes AM, Walter J, Segal E, Spector TD. Role of the gut microbiota in nutrition and health. BMJ. 2018 Jun 13;361:k2179. doi: 10.1136/bmj.k2179.

Li RJ, Liu Y, Liu HQ, Li J. Ketogenic diets and protective mechanisms in epilepsy, metabolic disorders, cancer, neuronal loss, and muscle and nerve degeneration. J Food Biochem. 2020 Jan 14:e13140. doi: 10.1111/jfbc.13140. [Epub ahead of print]

Clark WF, Sontrop JM, Huang S-H, et al. Effect of coaching to increase water intake on kidney function decline in adults with chronic kidney disease: the CKD wit randomized clinical trial. JAMA 2018;319:1870–9. 10.1001/jama.2018.4930

Evangelidis N1,2, Craig J2,3, Bauman A4, Manera K4,2, Saglimbene V4,2, Tong A4,2.Lifestyle behaviour change for preventing the progression of chronic kidney disease: a systematic review. BMJ Open. 2019 Oct 28;9(10):e031625. doi: 10.1136/bmjopen-2019-031625.

Reviewer 3 Report

The topic of this manuscript is of interest and adds to the field. The manuscript is generally comprehensible; however, a review from a native English speaker could improve this. For instance:
1) Sentence 1 in the abstract should read "have been shown" not "have been showed" and compare is spelled wrong.
2) "Specialists who work with patients, suffer from obesity..." Should be suffering.

Title
It would be helpful to include the study design in the title. For instance, title: a parallel group intervention.

Abstract
Sufficient detail is provided to understand the intention of the manuscript.

Keywords
I suggest that you add weight loss.

Introduction
The introduction provides sufficient background information. However, I do not feel that I have an understanding of the design of your study (what are the diets in the arms). A brief introduction when they are introduced would be helpful. It would also be helpful to explain how these were chosen/their relation to the literature/previous studies. Some of this is in the Discussion instead. We are missing the why of this study really, which should be in the Introduction.

Line 49 Patient first language is preferable. Should say adults with obesity, not obese adults. This highlights the disease as a state, not a defining characteristic of the individual.

Methods
The study seems to be appropriately designed and implements.

Subjects: "None of them had any previous experience with reductive diets before
115 this study." That seems hard to believe. Never?

Results
Tables are appropriately designed

Limitations
This should be made more clear in the manuscript.

A cross-over design could strengthen these findings and should be considered for future replication/confirmation studies.

Author Response

The topic of this manuscript is of interest and adds to the field. The manuscript is generally comprehensible; however, a review from a native English speaker could improve this. For instance:
1) Sentence 1 in the abstract should read "have been shown" not "have been showed" and compare is spelled wrong.
2) "Specialists who work with patients, suffer from obesity..." Should be suffering.

Answer: Thank you for this checking, we have improved above mentioned parts of manuscript and let it corrected by native speaker.

Title
It would be helpful to include the study design in the title. For instance, title: a parallel group intervention.

Answer: As rewiever requested we changed the title

The effects of a parallel group intervention by low energy moderate carbohydrate (MCD) and mixed (MixD) diet on HDL-C and LDL-C concentration in middle aged men.

Abstract
Sufficient detail is provided to understand the intention of the manuscript.

Answer: Thank you for this evaluation.

Keywords
I suggest that you add weight loss.

Answer: Done as requested

Introduction
The introduction provides sufficient background information. However, I do not feel that I have an understanding of the design of your study (what are the diets in the arms). A brief introduction when they are introduced would be helpful. It would also be helpful to explain how these were chosen/their relation to the literature/previous studies. Some of this is in the Discussion instead. We are missing the why of this study really, which should be in the Introduction.

Answer: As suggested by the reviewer, we completed the introduction of additional sentences in which we explained the purpose of research.

Line 49 Patient first language is preferable. Should say adults with obesity, not obese adults. This highlights the disease as a state, not a defining characteristic of the individual.

Answer: Dane as requested

Methods
The study seems to be appropriately designed and implements.

Answer: Thank you for this point, we little improve the description of the test and added some limitation be more clear.

Subjects: "None of them had any previous experience with reductive diets before
115 this study." That seems hard to believe. Never?

Answer: The sentence was corrected to less raw statement.

Results
Tables are appropriately designed

Limitations
This should be made more clear in the manuscript.

Answer: We have added the limitation part.

A cross-over design could strengthen these findings and should be considered for future replication/confirmation studies.

Answer: Thank you for this comment, we try to run also cross-over design in our studies. In this manuscript we have add the issue into limitation statement.

Round 2

Reviewer 1 Report

This is a revised manuscript as for a randomized controlled study to compare MCD, MixD, and CD. Although it was improved, it was not sufficient.

Major

Why did author change their statistical method? I only asked the reason why authors used a two-way repeated measure ANOVA. In this revised manuscript, authors used one-way repeated measure ANOVA. Furthermore, authors changed post-hoc analysis method from Bonferroni to Tukey. I cannot help recognizing authors statistical analysis as incorrect Authors improved their references. But it was not sufficient. For example, authors described “In the diet of modern Europe, the carbohydrate content ranges between 45 and 60% of all calories consumed” in lines 39-40. In their reference 7, carbohydrate content ranges between 45 and 60% is described as reference intake (p37, Table 3). On the other hand, in reference 7, consumed values have very big range 31.9(5%ile)-63.2%(95%ile) (p70, Annex 3B). Authors description is incorrect.

3.Authors described as below;

Lines 129. RMR was measured at the beginning of the experiment.

Lines 128. TDEE was calculated according to the equation TDEE = AF x RMR.

Lines 131. AF = 1.2

Lines 127. In MCD and MixD group, caloric consumption was 20% fewer calories than TDEE.

According to these descriptions, I can calculate as below.

TDI in MCD = 1.2 x RMR x 0.8 = 1.2 x 1736 x 0.8 = 1666.56 (different from 2170)

TDI in MixD = 1.2 x RMR x 0.8 = 1.2 x 1752 x 0.8 = 1681.92 (different from 2190)

Authors descriptions are doubtful.

Minor

In this trial, authors did not assess type of fatty acids. Authors should delete but not shorten the description as for type of fatty acids.

Author Response

Reviewer 1

Major

Why did author change their statistical method? I only asked the reason why authors used a two-way repeated measure ANOVA. In this revised manuscript, authors used one-way repeated measure ANOVA.

Answer:

We understand your previous comment “I feel one way factorial ANOVA is better, to compare the changes of parameters in three groups during intervention” as relevant and our statistician recommended the change for one way factorial (as recommended). Therefore, we perform this change which we believe is correct.

Furthermore, authors changed post-hoc analysis method from Bonferroni to Tukey. I cannot help recognizing authors statistical analysis as incorrect

Answer: We believe that for factorial ANOVA it is appropriate according to our statistician. We make this change according to ANOVA results change.

Authors improved their references. But it was not sufficient. For example, authors described “In the diet of modern Europe, the carbohydrate content ranges between 45 and 60% of all calories consumed” in lines 39-40. In their reference 7, carbohydrate content ranges between 45 and 60% is described as reference intake (p37, Table 3). On the other hand, in reference 7, consumed values have very big range 31.9(5%ile)-63.2%(95%ile) (p70, Annex 3B). Authors description is incorrect.

Answer: The sentences was corrected

3.Authors described as below;

Lines 129. RMR was measured at the beginning of the experiment.

Answer:

Lines 128. TDEE was calculated according to the equation TDEE = AF x RMR.

Lines 131. AF = 1.2

Lines 127. In MCD and MixD group, caloric consumption was 20% fewer calories than TDEE.

 According to these descriptions, I can calculate as below.

TDI in MCD = 1.2 x RMR x 0.8 = 1.2 x 1736 x 0.8 = 1666.56 (different from 2170)

TDI in MixD = 1.2 x RMR x 0.8 = 1.2 x 1752 x 0.8 = 1681.92 (different from 2190)

Authors descriptions are doubtful.

 Answer: Thank you for this fact checking. The reviewer was completely write. The value of RMR was wrong calculated. AF was not 1.2 but 1.4- like in firs manuscript version. We changed the RMR value and AF.

 In this trial, authors did not assess type of fatty acids. Authors should delete but not shorten the description as for type of fatty acids.

Answer: As reviewer  request this part was deleted.

Reviewer 2 Report

Authors have done great work in improving the manuscript. I have no further comments.

Author Response

Thank you again for helping us to improve our manuscript.